

# The beneficial effect of physical activity on cognitive function in community-dwelling older persons with locomotive syndrome

Misa Nakamura[1,2], Masakazu Imaoka[1,2], Hiroshi Hashizume[3,4], Fumie Tazaki[1,2], Mitsumasa Hida[1,2], Hidetoshi Nakao[1,2], Tomoko Omizu[5], Hideki Kanemoto[1,6] and Masatoshi Takeda[1]

[1] Cognitive Reserve Research Center, Osaka Kawasaki Rehabilitation University, Kaizuka, Osaka, Japan
[2] Department of Rehabilitation, Osaka Kawasaki Rehabilitation University, Kaizuka, Osaka, Japan
[3] Department of Orthopaedic Surgery, Wakayama Medical University, Wakayama, Wakayama, Japan
[4] School of Health and Nursing Science, Wakayama Medical University, Wakayama, Wakayama, Japan
[5] Department of Health and Medical Science, Kansai University of Welfare Sciences, Kashiwara, Osaka, Japan
[6] Department of Psychiatry, Osaka University Graduate School of Medicine, Suita, Osaka, Japan

Corresponding author
Misa Nakamura,
nakamuram@kawasakigakuen.ac.jp

## ABSTRACT

**Background:** Cognitive decline is closely related to motor decline. Locomotive syndrome (LS) is defined as a state associated with a high risk of requiring support because of locomotive organ disorders, and can be evaluated using a questionnaire. This study aimed to clarify the effectiveness of daily goal-targeted exercise on cognitive function in two different populations classified by scores on the Locomo 25 questionnaire.

**Methods:** Seventy community-dwelling older people who participated in a 13-week health class were divided into two populations based on Locomo 25 scores: <7 (non-LS) and ≥7 (LS). Participants were presented with a daily target steps and worked towards that goal. Cognitive function was evaluated using the Japanese version of Addenbrooke's Cognitive Examination-Revised (ACE-R). Average daily physical activity (exercise [Ex]) for 13 weeks was measured using a portable activity meter. Depression status was assessed using the Geriatric Depression Scale (GDS-15).

**Results:** No significant differences were observed in age, years of education, body mass index, smooth muscle mass index, GDS-15 scores, or ACE-R scores between the non-LS and LS populations. Multiple logistic regression analysis showed that Ex (odds ratio = 5.01, $p = 0.002$) for 13 weeks was significantly associated with increased cognitive function in the LS population. The Ex threshold for the increase in cognitive function based on receiver operating curve analysis was 2.29 metabolic equivalents of task (METs) × h (METs · h/day) ($p = 0.047$) in the LS population. After 13 weeks, ACE-R scores were significantly higher in the Ex ≥ 2.29 than in the Ex < 2.29 METs · h/day group ($p = 0.024$, $\eta_p^2 = 0.241$) in the LS population based on two-way analysis of covariance. Furthermore, a significant increase in the ACE-R memory domain was seen in the Ex ≥ 2.29 group ($p = 0.035$, $\eta_p^2 = 0.213$).

**Conclusions:** These results suggest that Ex ≥ 2.29 METs · h/day is important for improving cognitive function in LS populations.

# INTRODUCTION

In Japan, over 26% of the population was aged ≥ 65 years as of 2015, and by 2040, the proportion of older people (>75 years) is expected to surpass 20%; therefore, Japan is becoming a super-aging society (*Statistics Bureau of Japan, 2018*). Under these circumstances, to realize a society with healthy longevity, it is important to prevent the aged from falling into dysfunction or requiring nursing care. It has been reported that musculoskeletal diseases such as bone fractures and osteoporosis rank first, and dementia ranks third, in terms of causes for requiring long-term care; therefore, the prevention of these diseases is an urgent issue.

Locomotive syndrome (LS) was proposed by the Japanese Orthopaedic Association in 2007 as a state with a high risk of requiring health-care support or nursing care because of problems associated with locomotion. Some causes of LS include reduced muscle strength and balance associated with aging and locomotive pathologies such as sarcopenia, osteoarthritis, and osteoporosis (*Nakamura, 2008*; *Momoki et al., 2017*). LS has also been found to be closely related to frailty (*Yoshimura et al., 2019*). In recent years, we have analyzed cross-sectional studies on factors related to LS in community-dwelling older people and found that LS is associated with motor function (*Nakamura et al., 2015*), body mass index (BMI) (*Nakamura et al., 2016*), depression (*Nakamura et al., 2017a*), cognitive function (*Nakamura et al., 2017b*), and subjective oral dysfunction (*Nakamura et al., 2021*). Several studies on LS and motor function have been conducted (*Ikemoto & Arai, 2018*). The proportion of the population with LS (47 million) in Japan is estimated to be more than twice that with metabolic syndrome (20 million) (*Yoshimura et al., 2009*; *Ministry of Health, Labour and Welfare, 2016*). One of the methods used to evaluate the severity of LS is the Locomo 25 questionnaire, a 25-item LS screening tool that measures the degrees of physical pain, ability to carry out activities of daily living (ADL), sociality, and active living (*Nakamura, 2008*; *Wang et al., 2020*). A Locomo 25 score of ≥7 indicates LS (*Japanese Orthopedic Association, 2015*).

Adequate exercise and nutrition have been widely suggested as preventive measures for cognitive function decline. In addition, the state at which walking function and cognitive function are reduced at the pre-stage of dementia is called "motor cognitive risk syndrome" (*Verghese et al., 2014*), and the clinical significance of evaluating motor function in this way is being increasingly recognized. However, it remains unclear to what extent motor therapy is effective to prevent cognitive decline among people with reduced motor function. In addition, many health classes have introduced uniform exercise activities for their participants, but the results are somewhat unclear.

We previously identified a strong relationship between Locomo 25 and Mini-Mental State Examination scores, which are used to evaluate cognitive function, in older women living independently in the community: a Locomo 25 score of six points was the cutoff value for normal cognitive function, whereas a score of ≥6 indicated a higher risk for cognitive decline (*Nakamura et al., 2017b*). These findings suggest that cognitive decline would be prevented if musculoskeletal function could be recovered—*i.e.*, a reduction in the severity of LS—by some kind of intervention in the early stage using Locomo 25 as an index. It has been reported that LS-induced back pain improves with exercise (*Hashizume et al., 2014*), and that exercise interventions for osteoarthritis improve ADL (*Penninx et al., 2001*). Considering the findings from these previous studies, establishing preventive measures linking cognitive function with LS was considered to be potentially effective for reducing the need for long-term care.

Physical activity refers to all body movements by the skeletal muscles that expend energy, and includes whole-body endurance, muscular strength, balance ability and flexibility. By contrast, exercise is a planned, intentional, and continuous type of physical activity aimed at maintaining and improving physical strength. Physical activity is associated with mortality, in that higher levels of physical activity decrease the risk of death (*Gregg et al., 2003*). In addition, cognitive function has been reported to be closely related to motor function (*Maruya et al., 2018*), and both exercise and physical activity have been shown to be effective for increasing cognitive reserve for the prevention of dementia and for reducing brain damage and neuroinflammation (*Livingston et al., 2020*).

Given this background, the purposes of this study were to categorize community-dwelling older people based on the presence or absence of LS and to clarify the effectiveness of daily goal-targeted exercise on cognitive function in two different populations classified by Locomo 25 scores.

## MATERIALS & METHODS

### Participants

This study was conducted in Kaizuka city, Osaka Prefecture, from January to April 2019. The study participants were recruited by placing a recruitment leaflet in the local newspaper or posting it at the city hall. The exclusion criteria were: (1) <60 years of age; (2) previously diagnosed with dementia; (3) having a cardiac pacemaker; and (4) had stopped exercising on the advice of a physician. All applicants participated in a once-weekly 13-week exercise class co-hosted by Kaizuka city and Osaka Kawasaki Rehabilitation University. A previous study on the effects of a once-weekly 90-min walking program for community-dwelling older adults reported improved cognitive function after about 3 months (*Maki et al., 2012*); therefore, the intervention period in the present study was set to 13 weeks. The exercise class was conducted once a week (1 h per session) for 13 weeks by a certified physical therapist. The contents of the class included 15 min of intellectual tasks and 45 min of exercise tasks, such as soft gymnastics and light dancing. At the beginning of the study, a wireless activity meter with three-axis accelerometer sensors (AM510N; ACOS Co., Ltd., Nagano, Japan) was handed out to each person. This activity meter has been proven to have the same accuracy as that from other

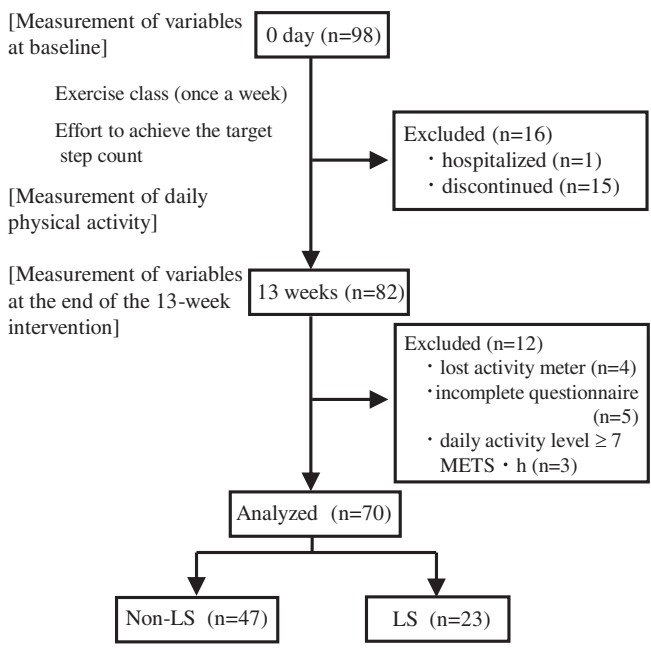

**Figure 1 Flowchart of the present study.** LS, Locomotive syndrome; METS, Metabolic equivalents of task.                               

companies (personal communication with ACOS Co.). The target values were 6,700 steps per day for men and 5,900 steps per day for women, which are the values proposed by the Ministry of Health, Labour and Welfare of Japan for the prevention of lifestyle-related diseases for people aged ≥ 70 years (*Ministry of Health, Labour and Welfare, Healthy Japan21, 2013*). The measurements described below were performed before participation (baseline) and after a 13-week intervention. A self-administered questionnaire on education history was also implemented.

In total, 98 individuals participated in the baseline measurement. During the 13-week study period, one patient was hospitalized and 15 discontinued. After 13 weeks, 82 participants remained, among whom, four who had lost their activity meter, five who had not answered the questionnaire completely, and three who had a daily activity level (exercise [Ex]) ≥ 7.0 (metabolic equivalents of task (METs) × h (METs · h) were excluded. Finally, data from 70 participants (20% men; mean age [standard deviation], 74.71 [5.26] years; age range, 63–91 years) were analyzed. A flowchart is shown in Fig. 1. This study was reviewed and approved by the Ethics Committee of Osaka Kawasaki Rehabilitation University (approval No.: OKRU29-A015), and carried out in accordance with the Declaration of Helsinki. Prior to the start of the survey, written informed consent was obtained from all participants. Participants with cognitive decline were permitted to ask their spouse, children, or other relatives to answer on their behalf.

## Measurement of variables

BMI was calculated as the weight in kilograms divided by the height in meters squared. Skeletal muscle mass index (SMI) was measured using a bioelectrical impedance analysis (BIA) device (InBody 270; InBody Japan Co., Ltd., Tokyo, Japan) at 20 and 1,000 kHz

while the participants were wearing normal indoor clothing without socks and shoes. All participants were instructed to grasp the handles of the BIA device and stand on electrodes contacting the bottoms of their feet. Physical activity was measured using a wireless activity meter (AM510N; ACOS Co., Ltd., Nagano, Japan) that the participants attached to their pants for 13 weeks and removed only when taking a bath. After 13 weeks, daily Ex (METs · h) for 13 weeks from the activity meter was read by a contactless integrated circuit card reader (Ferica RC-S380/S; Sony Co., Tokyo, Japan) and the data were transferred to an Excel file. The average Ex per day was calculated by dividing the total Ex for 13 weeks by 91 days.

## Evaluation of depression

Depression has been reported to be related to LS (*Nakamura et al., 2017b*) and cognition (*Byers & Yaffe, 2011*); therefore, we considered that depression could be a confounding factor. Depression status was assessed using the 15-item Geriatric Depression Scale (GDS-15), a widely used international instrument for depression screening in the general geriatric population with confirmed reliability and validity (*Demura et al., 2006*). Each item on the GDS-15 is measured using a yes/no questionnaire. All 15 items are scored as either 0 or 1, with the total score ranging from 0 to 15. Scores ≥ 5 are evaluated as depression (*Shiekh & Yesavage, 1986*).

## Evaluation of cognitive function

Cognitive function was evaluated using the Japanese version of Addenbrooke's Cognitive Examination-Revised (ACE-R) (*Mioshi et al., 2006*; *Hashimoto, 2010*), which is a revised version of the original ACE (*Mathuranath et al., 2000*). The ACE-R is strongly correlated with Clinical Dementia Rating (CDR) scale and is more accurate than the Mini-Mental State Examination, which is widely used internationally for the detection of mild cognitive impairment (MCI) (*Yoshida et al., 2011*). The ACE-R consists of the following five cognitive areas: attention and orientation (18 points), memory (26 points), language fluency (14 points), language (26 points), and visuospatial cognition (16 points). A perfect score on the ACE-R is 100 points (Table S1), while a score of ≥89 is normal, a score of 83–88 indicates MCI, and a score of ≤82 indicates dementia (*Yoshida et al., 2011*; *Yoshida et al., 2012*). The ACE-R was conducted in a one-on-one face-to-face manner. Version A was used at baseline, and version B at after 13 weeks. The classification, which is based on cognitive changes, was calculated by subtracting the baseline ACE-R score from that after 13 weeks. When this difference was >0, the participant was classified into the group with increased cognitive function, and when the difference was ≤0, the participant was classified into the group with non-increased cognitive function.

## Assessment of LS status

LS status was evaluated using the Locomo 25 score, which was previously known as the Geriatric Locomotive Function Scale-25 score. The Locomo 25 is a self-administered questionnaire composed of four questions about pain, 16 questions about ADL, three questions about social function, and two questions about mental health status during the

last month (*Seichi et al., 2012*). All 25 items are scored from 0 (no impairment) to four (severe impairment), with the total score ranging from 0 to 100. Higher scores indicate worse locomotive function, and a total score of ≥7 points is evaluated as LS (*Japanese Orthopedic Association, 2015*). Furthermore, in the LS risk classification, a Locomo 25 total score of 7–15 points indicates LS risk 1, 16–23 points LS risk 2 and ≥24 points LS risk 3 (*Japanese Orthopedic Association, 2015*). The validity of the Locomo 25 was confirmed by demonstrating a significant correlation with the European Quality of Life Scale-5 Dimensions questionnaire (*Seichi et al., 2012*).

## Statistical analysis

To compare numerical values between the two groups, the normality of distribution and homogeneity of variance were tested prior to comparison across groups. Student's *t*-test was used when assumptions of normal distribution and homogeneity of variance were fulfilled in both groups, and Welch's *t*-test was used when the assumption of normal distribution was met, but not the assumption of homogeneity of variance. When the data were non-normally distributed, the Wilcoxon signed-rank test was used. Pearson's chi-squared test was used to compare exercise habits between the non-LS and LS groups. To clarify the effect of Ex on cognitive improvement after the 13-week intervention in the Non-LS and LS groups, the odds ratios (ORs) of measurements for an ACE-R score of >0 were calculated using logistic regression analysis adjusted for age. Ex used as independent variables, and an ACE-R score of >0 (increased cognitive function) was used as dependent variable. The Ex threshold score for discriminating between the increased and non-increased cognitive function groups was evaluated using receiver operating curve (ROC) analysis. The ORs of the cutoff values for increased cognitive function according to Ex were calculated using multiple logistic regression analyses adjusted for age. Two-way analysis of covariance (ANCOVA) using age and baseline as covariates was carried out to compare ΔACE-R scores and ΔACE-R domain scores to determine the effect of Ex on outcomes in the LS group after the 13-week study period. Partial eta-squared ($\eta_p^2$) is the value of the sum of squares (SSA) divided by the SSA plus the sum of squared errors of prediction. The $\eta_p^2$ values describe an effect size of 0.01 as small, 0.06 as medium and 0.14 as large (*Cohen, 1988*). Two-way ANCOVA was conducted using SPSS Statistics software (version 26; IBM Corp., Armonk, NY), and other statistical analyses using JMP 14 (SAS Institute, Cary, NC). All statistical tests were two-tailed, and a significance level of 0.05 was used.

## RESULTS

The participants' age, years of education, BMI, SMI, GDS-15, ACE-R, Locomo 25 scores and exercise habits are shown in Table 1. No significant differences in age, years of education, BMI, SMI, or GDS-15 and ACE-R scores were found between the Non-LS and LS groups. Exercise habits before the start of the exercise class did not significantly differ between the two groups ($p = 0.804$) (Table 1).

Table 2 shows a comparison of characteristics and changes in measurements from baseline to 13 weeks and physical activity levels for 13 weeks between non-increased

**Table 1 Comparison of measurements at baseline and physical activity levels during the 13-week study period and between the non-LS and LS populations.**

|  | All | Non-LS | LS | p |
|---|---|---|---|---|
| N (% men) | 70 (20.0%) | 47 (23.4%) | 23 (13.0%) |  |
| Age (y) | 74.71 (5.26) | 73.94 (5.12) | 76.30 (5.30) | 0.077 |
| Years of education (y) | 12.27 (2.35) | 12.43 (2.33) | 11.96 (2.40) | 0.436 |
| BMI (kg/m$^2$) | 22.88 (2.89) | 22.80 (2.77) | 23.03 (3.17) | 0.765 |
| SMI (kg/m$^2$) | 6.00 (0.90) | 6.07 (0.89) | 5.85 (0.92) | 0.333 |
| GDS-15 (points) | 2.97 (2.54) | 2.64 (2.63) | 3.65 (2.25) | 0.118 |
| ACE-R score (points) | 90.37 (8.15) | 91.28 (7.29) | 88.52 (9.58) | 0.186 |
| Locomo 25 score (points) | 7.63 (8.68) | 2.74 (1.80) | 17.61 (8.63) | <0.0001 |
| Exercise 7 days/week | 18.6% | 21.3% | 13.0% | 0.804 |
| 5–6 days/week | 4.3% | 4.3% | 4.3% |  |
| 1–4 days/week | 60.0% | 59.6% | 60.9% |  |
| 0 days/week | 17.1% | 14.9% | 21.7% |  |

Notes:
All values are shown as mean (standard deviation) or prevalence (percentage).
Non-LS, Non-locomotive syndrome; LS, Locomotive syndrome; BMI, Body mass index; SMI, Skeletal muscle mass index; GDS-15, Geriatric Depression Scale-15; ACE-R, Addenbrooke's Cognitive Examination-Revised.

**Table 2 Comparison of group characteristics and changes in values during the 13-week study period.**

|  | Non-LS | | LS | |
|---|---|---|---|---|
|  | ΔACE-R score ≤ 0 | ΔACE-R score > 0 | ΔACE-R score ≤ 0 | ΔACE-R score > 0 |
| N (% men) | 24 (20.83%) | 23 (26.09%) | 13 (7.69%) | 10 (20.00%) |
| Age (y) | 74.46 (4.81) | 73.39 (5.48) | 74.62 (4.31) | 78.50 (5.87) |
| Years of education (y) | 12.13 (2.47) | 12.74 (2.18) | 12.31 (2.32) | 11.50 (2.55) |
| ΔBMI (kg/m$^2$) | −0.19 (0.48) | −0.29 (0.29) | −0.22 (0.57) | −0.32 (0.53) |
| ΔSMI (kg/m$^2$) | 0.13 (0.15) | 0.09 (0.12) | 0.05 (0.13) | 0.07 (0.16) |
| ΔGDS-15 (points) | −0.54 (1.14) | −0.17 (2.08) | −0.92 (2.25) | −0.50 (1.27) |
| ΔACE-R score (points) | −3.33 (3.27) | 3.09 (1.90)[†] | −3.92 (3.62) | 4.30 (2.87)[‡] |
| ΔLocomo 25 score (points) | 1.88 (6.44) | 0.48 (2.33) | 2.15 (9.10) | −1.50 (8.36) |
| Ex (METs · h/day) | 2.56 (1.31) | 2.87 (1.36) | 1.72 (0.71) | 2.98 (1.49)[‡] |

Notes:
All values are shown as mean (standard deviation) or prevalence (percentage).
[†] $p < 0.05$ (vs. ACE-R score ≤ 0 in the non-LS group).
[‡] $p < 0.05$ (vs. ACE-R score ≤ 0 in the LS group).
Non-LS, Non-locomotive syndrome; LS, Locomotive syndrome; Δ, Change from baseline to 13 weeks; BMI, Body mass index; SMI, Skeletal muscle mass index; GDS-15, Geriatric Depression Scale-15; ACE-R, Addenbrooke's Cognitive Examination-Revised; Ex, Daily exercise for 13 weeks; METs, Metabolic equivalents of task.

cognitive function (ΔACE-R ≤ 0) and increased cognitive function (ΔACE-R > 0) in the non-LS and LS populations. In the LS population, Ex was significantly higher for increased compared with non-increased cognitive function; however, no significant differences were observed for any changes in the non-LS population (Table 2).

Table 3 shows the results of logistic regression analysis adjusted for age and the ORs of each measurement, including GDS-15 scores and Ex in the non-LS and LS populations. The results showed that Ex was significantly associated with increased cognitive function

**Table 3 Odds ratios of characteristics for increased cognitive function[†] during the 13-week study period.**

|  | Non-LS | | | LS | | |
|---|---|---|---|---|---|---|
|  | OR | 95% CI | p | OR | 95% CI | p |
| Ex (METs · h/day) | 1.18 | [0.76–1.85] | 0.460 | 5.01 | [1.30–19.24] | 0.002 |

Notes:
Multiple logistic regression analysis adjusted for age.
[†] Increased cognitive function; ΔACE-R score > 0; Non-LS, Non-locomotive syndrome; LS, Locomotive syndrome; Ex, Daily exercise; METs, Metabolic equivalents of task.

**Table 4 Threshold of Ex values for increased cognitive function[†] and odds ratios (ORs) for increased cognitive function according to Ex in the LS population.**

| Threshold Ex value (METs · h/day)[‡] | Area under the curve | Sensitivity (%) | Specificity (%) | p | Ex (METs · h/day) | OR[§] | 95% CI | p |
|---|---|---|---|---|---|---|---|---|
| 2.29 | 0.808 | 80.00 | 84.62 | **0.047** | <2.29 | 1 | [2.11–228.12] | 0.010 |
|  |  |  |  |  | ≥2.29 | 21.94 |  |  |

Notes:
[‡] Receiver operating characteristic curve analysis was performed.
[§] Multiple logistic regression analysis adjusted for age was performed.
[†] Increased cognitive function; ΔACE-R score > 0; LS, Locomotive syndrome; Ex, Exercise; METs, Metabolic equivalents of task; CI, Confidence interval.

**Table 5 Comparison of Δ ACE-R scores between the two Ex groups in the LS population.**

| Ex (METs · h/day) | ACE-R at baseline | ACE-R after 13 weeks | ΔACE-R | F | p | $\eta_p^2$ |
|---|---|---|---|---|---|---|
| Ex < 2.29 | 89.15 (11.31)[†] | 86.46 (10.52) | –2.69 (4.96) | 6.022 | 0.024 | 0.241 |
| Ex ≥ 2.29 | 87.70 (7.24) | 90.40 (8.17) | 2.70 (4.14) |  |  |  |

Notes:
Two-way analysis of covariance using baseline and age as the covariates.
[†] Values are shown as means (standard deviation).
LS, Locomotive syndrome; Ex, Exercise; METs, Metabolic equivalents of task; ACE-R, Addenbrooke's Cognitive Examination-Revised; $\eta_p^2$, Partial eta-squared.

(OR = 5.01, 95% confidence interval CI = [1.30–19.24]; $p$ = 0.002). However, no such significant association was detected in the non-LS population (Table 3).

ROC analysis was conducted in regard to Ex in the LS population. A threshold for discriminating between the non-increased and increased cognitive function groups was identified as follows. A high area under the ROC curve (AUC) value (0.808, $p$ = 0.047) was observed with an Ex threshold of 2.29 METs · h/day in the LS population (Table 4). The ORs for the prevalence of increased cognitive function according to the threshold Ex values in the LS population are shown in Table 4. In the LS population, the high Ex group (≥2.29 METs · h/day) showed increased cognitive function, with an OR of 21.94 (95% CI [2.11–228.12], $p$ = 0.010) as determined by multiple logistic regression analysis adjusted for age (Table 4).

Table 5 shows a comparison between the two Ex groups among the LS population by two-way ANCOVA using age and baseline as the covariates. The ΔACE-R scores were significantly higher in the Ex ≥ 2.29 than in the Ex < 2.29 METs · h/day group, with a large effect size ($p$ = 0.024, $\eta_p^2$ = 0.241) (Table 5).

Table 6 Comparison of Δ ACE-R domain scores between the two Ex groups in the LS population.

| Ex (METs · h/day) | ACE-R at baseline | ACE-R after 13 weeks | ΔACE-R | $F$ | $p$ | $\eta_p^2$ |
|---|---|---|---|---|---|---|
| Orientation/Attention (points) | | | | | | |
| Ex < 2.29 | 17.46[†] (1.13) | 17.54 (1.39) | 0.08 (1.66) | 0.020 | 0.888 | 0.001 |
| Ex ≥ 2.29 | 17.20 (0.79) | 17.40 (0.84) | 0.20 (1.23) | | | |
| Memory (points) | | | | | | |
| Ex < 2.29 | 21.08 (5.55) | 18.62 (5.12) | −2.46 (4.79) | 5.134 | **0.035** | 0.213 |
| Ex ≥ 2.29 | 18.60 (5.17) | 20.70 (5.08) | 2.10 (2.60) | | | |
| Verbal fluency (points) | | | | | | |
| Ex < 2.29 | 11.85 (2.51) | 10.92 (3.64) | −0.92 (2.62) | 2.244 | 0.151 | 0.106 |
| Ex ≥ 2.29 | 12.60 (1.51) | 12.80 (1.69) | 0.20 (2.62) | | | |
| Language (points) | | | | | | |
| Ex < 2.29 | 24.69 (2.75) | 23.92 (2.46) | −0.77 (1.30) | 0.063 | 0.805 | 0.003 |
| Ex ≥ 2.29 | 25.30 (1.89) | 24.60 (2.55) | −0.70 (1.34) | | | |
| Visuospatial (points) | | | | | | |
| Ex < 2.29 | 14.08 (1.19) | 15.46 (0.88) | 1.38 (1.26) | 2.139 | 0.160 | 0.101 |
| Ex ≥ 2.29 | 14.00 (1.41) | 14.80 (1.62) | 0.80 (0.92) | | | |

Notes:
Two-way analysis of covariance using baseline and age as the covariates.
[†] Values are shown as means (standard deviation).
LS, Locomotive syndrome; Ex, Exercise; METs, Metabolic equivalents of task; ACE-R, Addenbrooke's Cognitive Examination-Revised; $\eta_p^2$, Partial eta-squared.

Table 6 shows a comparison of changes from baseline to 13 weeks in ACE-R domain scores between the Ex < 2.29 and Ex ≥ 2.29 METs · h/day groups with LS by two-way ANCOVA using age and baseline as the covariates. Memory scores were significantly higher in the Ex ≥ 2.29 than in the Ex < 2.29 METs · h/day group, with a large effect size ($p = 0.035$, $\eta_p^2 = 0.213$) (Table 6).

## DISCUSSION

The purposes of this study were to categorize community-dwelling older people based on the presence or absence of LS and to clarify the effectiveness of daily goal-targeted exercise on cognitive function in LS and non-LS groups. The results revealed a relationship between physical activity and cognitive function in the LS population; however, no such relationship was found in the non-LS population. The increased cognitive function during the 13-week study period was associated with higher average Ex in the LS population, and the ROC analysis showed a threshold of 2.29 METs · h/day. On the other hand, no significant relationship was found between Ex and cognitive function in the non-LS population. When the LS population was classified into Ex < 2.29 and Ex ≥ 2.29 METs · h/day groups, a significant increase in ACE-R scores and a decrease in Locomo 25 scores were observed in the Ex ≥ 2.29 group. As exercise habits before the start of the intervention did not significantly differ between the non-LS and LS groups ($p = 0.804$; Table 1), the difference found in the amount of physical activity in this study was considered to be the result of the effort of each participant in aiming to achieve the target

step count presented at the beginning of the study. From these results, it is considered that Ex ≥ 2.29 METs · h/day was effective in improving LS and restoring cognitive function in the LS population.

There have been many reports on the relationship between exercise/physical activity and events related to cognitive function (*Livingston et al., 2020*). A clinical study reported a correlation between low levels of motor function and high amyloid β deposition (*Tian et al., 2017*). Cohort studies have reported that a high level of physical activity reduces the risk of cognitive decline and Alzheimer's disease (AD) (*Warburton et al., 2010*; *Plassman et al., 2010*; *Wennberg et al., 2017*). Thus, there are two aspects of a high level of physical activity: one suppresses or delays the development of AD pathology, and the other increases other cognitive functions independently of AD pathology, *i.e.*, it compensates for impaired cognitive function and promotes high brain plasticity (*Buchman et al., 2019*). It is speculated that these mechanisms are involved in the relationship between physical activity and cognitive function identified in this study.

Although depression has been reported to be related to LS (*Nakamura et al., 2017a*) and cognition (*Byers & Yaffe, 2011*), in the present study, no relationship was found between GDS-15 scores at baseline or LS status and cognitive changes after 13 weeks, as shown in Tables 1 and 2. The reason for this was considered to be that the overall GDS-15 scores were low and cognitive function high in the participants in the present study.

Table 2 shows that the Locomo 25 score tended to decrease during the 13-week study period in the increased cognitive function group in the LS population, although no statistically significant difference was found. This result suggests that Ex ≥ 2.29 METs · h/day reduces the level of LS. Locomo 25 mainly includes questions about the degrees of physical pain, ADL, social function, and mental health. According to a recent survey by the Ministry of Health, Labour and Welfare of Japan (2016), chronic pain in the locomotive organs is severe, and usually at sites necessary for movement, such as the hip and lower limb joints, so pain is strongly associated with LS. Physical activity has been reported to be associated with body pain (*Naugle et al., 2017*), ADL (*Wearing et al., 2020*), and mental health (*Chekroud et al., 2018*), and exercise has been reported to improve low back pain (*Hashizume et al., 2014*) and ADL (*Penninx et al., 2001*). Therefore, it was considered that the decrease in Locomo 25 scores in the Ex ≥ 2.29 METs · h/day group with LS was due to an increase in physical activity, a decrease in pain, and an increase in the ability to carry out ADL.

According to the *Ministry of Health, Labour and Welfare's (2013)* "Physical Activity Standards for Health Promotion 2013", Ex ≥ 10 METs · h/week should be maintained regardless of strength for health promotion among older people aged ≥65 years. However, the results of the present study suggest that Ex ≥ 16 METs · h/week is needed to improve cognitive function; 10 METs · h/week had no effect on improving cognitive function in people with LS.

In the present study, a significant increase was observed in the memory score domain of the ACE-R in the group with Ex ≥ 2.29 METs · h/day in the LS population. It has been reported that moderate physical activity is positively correlated with hippocampal capacity,

as is physical activity and memory (*Makizako et al., 2015*). It has also been reported that smaller hippocampal volume in the healthy aged is associated with severe acute and chronic pain (*Zimmerman et al., 2009*). Furthermore, a longitudinal cohort of older people found that persistent pain is associated with decreased memory and an increased likelihood of dementia (*Whitlock et al., 2017*). The results of the present study suggest that the LS population may have increased memory owing to increased physical activity and pain relief.

In this study, no relationship was found between physical activity and cognitive function in the non-LS population. Since non-LS groups have potentially high physical function, it may be necessary to set a high Ex target value for physical activity to affect cognitive function. Furthermore, factors other than physical activity, such as dietary patterns (*Gillette-Guyonnet, Secher & Vellas, 2013*; *Gu & Scaneas, 2011*), intellectual activity (*Wang et al., 2002*), and sleep patterns (*Ma et al., 2020*), may be involved in the recovery of cognitive function, so these should be investigated in the future.

This study did have several limitations. First, the number of study participants ($n = 70$) was small. Second, the criteria for LS in terms of Locomo 25 scores have two levels: $\geq 7$ points and $\geq 16$ points. However, in this study, few people in the target population had a Locomo 25 score of $\geq 16$ points, so a score of $\geq 7$ was considered to indicate LS. In the future, exercise intervention times can be expected to be identified more accurately by conducting surveys on more participants and categorizing them into more severe and milder groups in the LS population. Third, it was not possible to separate clearly physical activity from exercise, so the content of the measured physical activity was not clear. Fourth, individuals who had been diagnosed with dementia in the hospital were excluded from the analysis in this study, but some were in the dementia category in the ACE-R assessment. In the future, it will be necessary to use a system in which the number of people in the cognitive function category is equalized at baseline between the two groups. In addition, since minimal clinically important differences are unclear in the ACE-R assessment, the differences in ACE-R scores before and after the intervention were used as an index of cognitive change. It will therefore be necessary to consider criteria for "ACE-R improvements" in the future.

## CONCLUSIONS

In this study, community-dwelling older people were categorized based on the presence or absence of LS as classified by Locomo 25 scores, and the effect of daily goal-targeted exercise on cognitive function was clarified in the two groups. The results revealed a relationship between physical activity and cognitive function in the LS population; however, no such relationship was found in the non-LS population. ACE-R scores were significantly higher in the Ex $\geq 2.29$ than in the Ex $< 2.29$ METs · h/day group in the LS population. Furthermore, a significant increase in the ACE-R memory domain was seen in the Ex $\geq 2.29$ group. These results suggest that EX $2.29 \geq$ METs · h/day is important for improving cognitive function in LS populations with decreased locomotor function. In the future, it will be important to analyze the effectiveness of exercise together with detailed LS status and cognitive function categories under greater control.

## ACKNOWLEDGEMENTS

The authors wish to thank Ms. Kaori Hamamura, Ms. Hiroko Fujiwara, and Mr. Yuji Tsukuda of the Kaizuka City Office, Welfare Department Elderly Care section, and Ms. Ritsuko Tanaka and the many students at Osaka Kawasaki Rehabilitation University for their assistance with examinations and measurements.

### Funding

This work was supported by JSPS KAKENHI (Grant No. 18K10800) and The Research Foundation for Dementia of Osaka. The funders had no role in study design, data collection and analysis, decision to publish, or preparation of the manuscript.

### Grant Disclosures

The following grant information was disclosed by the authors:
JSPS KAKENHI: 18K10800.
The Research Foundation for Dementia of Osaka.

### Competing Interests

The authors declare that they have no competing interests.

### Author Contributions

- Misa Nakamura conceived and designed the experiments, performed the experiments, analyzed the data, prepared figures and/or tables, authored or reviewed drafts of the paper, and approved the final draft.
- Masakazu Imaoka performed the experiments, prepared figures and/or tables, and approved the final draft.
- Hiroshi Hashizume analyzed the data, authored or reviewed drafts of the paper, and approved the final draft.
- Fumie Tazaki performed the experiments, prepared figures and/or tables, and approved the final draft.
- Mitsumasa Hida performed the experiments, prepared figures and/or tables, and approved the final draft.
- Hidetoshi Nakao performed the experiments, prepared figures and/or tables, and approved the final draft.
- Tomoko Omizu performed the experiments, prepared figures and/or tables, and approved the final draft.
- Hideki Kanemoto performed the experiments, authored or reviewed drafts of the paper, and approved the final draft.
- Masatoshi Takeda conceived and designed the experiments, authored or reviewed drafts of the paper, and approved the final draft.

## Human Ethics

The following information was supplied relating to ethical approvals (*i.e.*, approving body and any reference numbers):

This study was reviewed and approved by the Ethics Committee of Osaka Kawasaki Rehabilitation University (OKRU29-A015).

## Data Availability

The raw measurements are available in the Supplemental Files.

## Supplemental Information

Supplemental information for this article can be found online at http://dx.doi.org/10.7717/peerj.12292#supplemental-information.

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
