# Peer review of "The beneficial effect of physical activity on cognitive function in community-dwelling older persons with locomotive syndrome"

_PeerJ, doi:10.7717/peerj.12292_

## Round 0.1 · original submission · Minor Revisions

I have also read this manuscript, and I think this manuscript contains very interesting and useful data. Two expert reviewers have both acknowledged the interestingness and given suggestions to authors to improve this manuscript.

And most of all, we apologize for the significant delay in responding due to editorial issues.

·

Basic reporting

This paper is written in a clear and concise manner, and is generally easy to follow.

One important difficulty for me while reading the text was the differentiation between excercise and physical activity. On line 94-97, the authors seem to make a clear distinction between the two, but after this both terms are used interchangeably. If excercise is actually defined as "planned, intentional, ....physical activity" it may be questioned how this was measured in this research. As far as I know an activity tracker tracks all PA throughout the day and night and does not provide details on planned/intentional PA. Consider adding this as a limitation or making this differentiation more clear.

Second, while reading the discussion it seemed like a lot of new information/literature references were mentioned that were not yet presented in the introduction. For instance the paragraph on line 258-271 - i would suggest to mention some of these details/insights already in the introduction and keep it shorter in the discussion (keep line 266-271).

Line 247-248: authors should be careful in not mixing up light physical activity and sedentary lifestyle, which are two different and independent parameters. There is a difference between a person who is sedentary and a person who is physically inactive. Being 'physically inactive' means not doing enough physical activity (in other words, not meeting the physical activity guidelines ). However, being 'sedentary' means sitting or lying down for long periods.

Experimental design

My biggest concern is that the actual problem statement / research aims were not sufficiently clear to me after reading the introduction. I would suggest to revise this and make sure the research questions are linked to existing gaps in the current literature.
All necessary material are in there, but I feel like it needs some restructering and better linking of the different parts, systematically building up to stating the aim of this work at the end of the introduction.
For instance:
- line 57-58, maybe try to link this some more to other research or numbers showing that LS is an important condition to attend to
- line 74-77, I would suggest to revise the wording and motivate why it is important to examine this based on previous evidence or literature (gaps) rather than saying "we thought it would be effective".


Second, I think more details are needed on the measures included to assess the main variables. Especially for the activity registration, LS status, and measure of cognitive function.
- line 143-144: please provide more details on this procedure of data extraction, what is an IC card reader an how is it used, which data can be extracted from this device and how were they processed (i.e. how were raw data transformed in average Ex data?)
- line 154-161: please provide more details on the specific subdomains of cognitive function measured by this scale. Later, in Table 6, these are mentioned but seem to come out of the blue and need more information/details to explain how these were measured in the methods section. Consider adding example items.
- line 165-167: it would be useful to have a better idea on the actual questions that are part of this survey. Consider to provide them in a table or give one example item per domain.

Finally, it is not clear why depression status was measured as this is not motivated in the introduction or research questions, and authors do not discuss it in the results section or discussion. I wonder if authors could tell me why this was the case and how this is an essential variable.

Validity of the findings

Authors could consider to discuss potential threats to internal validity of their findings. That is, specificy the experimental design and how this affects causality inferences.
Specifically, as no control group is includeid, it is not possible to say if cognitive improvements were caused by the excercise program or potentially happened spontaneously over time or caused by other factors. Authors should consider to discuss this limitation.

Also, necessary information is missing on how participants were recruited. Authors could consider to discuss their sampling strategy and additionally mention how this relates to and may hinder the external validity of their findings. I'm specifically concerned about the potential big differences in cognitive function at baseline in this group. In the method section it is mentioned that people with severe congitive issues were helped by a family member, but this makes me wonder how much people differed in terms of cognitive function at baseline (also within the LS or non-LS group) and if this might have influenced effects/improvements. I'm concerned about potential ceiling effects in those already functioning well at baseline (how much improvement is possible over 13 weeks in those doing well?). People with dementia were excluded, but what about people already showing mild cognitive impairments at baseline?

In general, in the discussion, I miss clear links with the original research questions. Please consider to start the discussion with a short recap of the aims and link the discussion of the results to them throughout the text. Same holds for the part on statistical analyses (line 172-193).

I think authors should be careful with using too much speculation in the discussion. Especially the interpretations about the links with pain sometimes seem far-fetched, as no results are reported on pain levels in this group. If this is such an important outcome, I would suggest to report specific results of the pain item and analyze how this relates to excercise and cognitive function instead of making assumptions based on a composite Locomo score which reflects more domains than pain alone...

Additional comments

I think this is a very interesting paper reporting on the links between physical activity and cognitive function in older adults with locomotive syndrom. This paper is well-writen with and generally easy to read. In times of a growing older population with rates of dementia rapidly increasing, it is worth looking how lifestyle behaviors such as physical acitivity might help to prevent this. Overall, findings will contribute to a growing body of literature on this topic. Yet, I have outlined some minor and major comments and concerns that - if addressed - might potentially help to improve the structure and flow of the paper.

Reviewer 2 ·

Basic reporting

The title, abstract, introduction, methods, results and discussion are appropriate for the content of the text. Furthermore, the article is well constructed, the experiments are well conducted, and analysis is well performed. The figures are relevant, high quality, well labelled and described.

Experimental design

The angle of the research is original and the research is within the scope of the journal. Research question is well defined, relevant and meaningful. The overview and their proposal for a more suitable technology is highly technical, ethical and logistical.

Validity of the findings

The introduction is comprehensive. The findings are meaningful. Odds ratio is an appropriate measure to estimate the association between an exposure and an outcome in this study. The conclusions are well stated and relevant to the research questions.

Additional comments

This observational study investigated the efficacy of daily physical activity on cognitive function in older
persons with locomotive syndrome (LS). The authors recruited 70 community-dwelling older people with and without locomotive syndrome and measured their physical activity. The authors found that daily physical activity of 2.29 METs·h/day or higher was important for improving cognitive function in LS populations.



Editorial Criteria
BASIC REPORTING
The title, abstract, introduction, methods, results and discussion are appropriate for the content of the text. Furthermore, the article is well constructed, the experiments are well conducted, and analysis is well performed. The figures are relevant, high quality, well labelled and described.
EXPERIMENTAL DESIGN
The angle of the research is original and the research is within the scope of the journal. Research question is well defined, relevant and meaningful. The overview and their proposal for a more suitable technology is highly technical, ethical and logistical.
VALIDITY OF THE FINDINGS
The introduction is comprehensive. The findings are meaningful. Odds ratio is an appropriate measure to estimate the association between an exposure and an outcome in this study. The conclusions are well stated and relevant to the research questions.

Overall, I think this observational study is novel and will be of interest to others in the community of cognitive disease and depression research. This study is a case control study, whose major strength is that it can provide rapid and useful information on a smaller sample size. This research paper does an excellent job demonstrating the significant difference between the group of LS participants with and without daily physical activity. In general, the work is convincing except some major and minor comments below:


Major Comments:

Due to the long course of follow-up, was there any loss to follow-up for the 70 participants? If so, how did the authors deal with this major challenge and potential source of bias?

Was the reassessment of exposure status required in this study? Do you think it is needed?




Minor Comments:
I’m wondering how the period of 13 weeks was decided?

How did the author decide when to evaluate depression and cognitive function? What if the LS has a long latency period between an exposure of physical activity and outcome?

I recommend adding a flowchart to demonstrate the study design.

Annotated reviews are not available for download in order to protect the identity of reviewers who chose to remain anonymous.

---

## Round 0.2 · Minor Revisions

The corrections of this version by authors were adequate. Some more corrections have been requested by Reviewer 1. I recommend that the authors respond to them and resubmit a new version of the manuscript.

·

Basic reporting

Small commment. The authors mention MCI on p.6 (line 168) but as this is the first time to use this acronym, this should be written in full (Mild Cognitive Impairment).

Experimental design

Please provide validity and reliability information with each of the measures described in the Methods sections. This is missing for the activity meter, GDS-15 and ACE-R.

Validity of the findings

Although authors have made considerable changes to the Discussion, I'm still missing links with original research questions / aims / hypotheses and conclusions about wheter or not expectations are (not) confirmed by the findings. I still believe that this section would benefit from this additional information. For example: "This study aimed to investigate XX, this was (not) confirmed by the findings..."

Also, I wonder if authors could elaborate some more about the absence of a link between exercise and cognitive function in the non-LS group. One could expect that physical activity has a positive influence on cognitive function in this population as well. Could authors elaborate on the "other factors" that might improve cognitive function in this group on p. 9, line 305. Additionally, could authors elaborate on potential other reasons for the absence of this link?

On p.9, line 314: it is strange to say that the content of the measured physical activity was "a little unclear". Please revise wording. Also, could authors elaborate on the effect this might have had on the interpretation of your results.

On p.9, line 316: it is not clear what is suggested with "equalization of cognitive function", could authors make this statement more clear?

Finally, the conclusion at the end of the paper is very brief. Consider to add information on the implications of the findings and more specific directions for future research here as well.

Additional comments

no comment

Reviewer 2 ·

Basic reporting

As for the second version, the content is more comprehensive. The whole article is well constructed, the experiments are well conducted, and analysis is well performed. The figures are relevant, high quality, well labelled and described.

Experimental design

The angle of the research is original and the research is within the scope of the journal. Research question is well defined, relevant and meaningful. The overview and their proposal for a more suitable technology is highly technical, ethical and logistical.

Validity of the findings

The introduction is comprehensive. The findings are meaningful. Odds ratio is an appropriate measure to estimate the association between an exposure and an outcome in this study. The conclusions are well stated and relevant to the research questions.

Additional comments

This observational study investigated the efficacy of daily physical activity on cognitive function in older persons with locomotive syndrome (LS). The authors recruited 70 community-dwelling older people with and without locomotive syndrome and measured their physical activity. The authors found that daily physical activity of 2.29 METs·h/day or higher was important for improving cognitive function in LS populations.


Major Comments:

Due to the long course of follow-up, was there any loss to follow-up for the 70 participants? If so, how did the authors deal with this major challenge and potential source of bias?

Response: This is a valuable point. At the start of the intervention, there were 98 people, but data from 28 people could not be used midway through the study (16 were unable to participate in the study for some reason, 9 had incomplete data, and 3 had a daily activity level 7.0 METs·h higher [lines 130–136]), so finally, data from 70 people were analyzed. The purpose of this study was to target people who were living independently in the community, not to impose particularly intense physical activities, so 70 people were considered to meet that purpose. In addition, individuals with high physical activity (7.0 METs·h) greatly exceeding the suggested target amount were excluded from the analysis.

Feedback: Thanks for the clarification. It totally makes sense to me.

Was the reassessment of exposure status required in this study? Do you think it is needed?

Response: Numerical values such as those for body composition parameters were measured using an instrument, so no reevaluation was necessary. For the evaluations of LS, ACE-R, and GDS-15, the errors due to differences in evaluators and the timing of responding to the questionnaire were small, so we considered the obtained results to be highly reliable. Thus, we think that no reevaluation is necessary.

Feedback: Thanks for the explanation, I agree with you that the reassessment is unnecessary in this case.




Minor Comments:
I’m wondering how the period of 13 weeks was decided?
Response: This decision was made with reference to a previous report on improvements in cognitive function in community-dwelling older after a 3-month exercise intervention (lines 114–117).
Feedback: Thanks for the information. I think it would be better if you could say sentences like “according to the previous study, we decided that ...”

How did the author decide when to evaluate depression and cognitive function? What if the LS has a long latency period between an exposure of physical activity and outcome? I recommend adding a flowchart to demonstrate the study design.
Response: Thank you for pointing this out. All assessments were made before the start of the intervention and after the 13-week intervention. The effects of physical activity were clarified by dividing the participants into non-LS and LS groups before the start of intervention and then comparing the evaluations between the two points. A flowchart is now shown in Figure 1.
Feedback: Thanks for the clarification. It would be more clear if the authors could add it to the discussion section, since readers may raise these kinds of questions. The flowchart looks good to me. Thanks for making the plot.

Annotated reviews are not available for download in order to protect the identity of reviewers who chose to remain anonymous.

---

## Round 0.3 · accepted · Accept

I would like to apologize for a long time for selecting the reviewers, and thank you for submitting your great work to PeerJ.